# Simple synthesis of massively parallel RNA microarrays via enzymatic conversion from DNA microarrays

Erika Schaudy ⬤ [1], Kathrin Hölz[1], Jory Lietard ⬤ [1] & Mark M. Somoza ⬤ [1,2,3 ✉]

RNA catalytic and binding interactions with proteins and small molecules are fundamental elements of cellular life processes as well as the basis for RNA therapeutics and molecular engineering. In the absence of quantitative predictive capacity for such bioaffinity interactions, high throughput experimental approaches are needed to sufficiently sample RNA sequence space. Here we report on a simple and highly accessible approach to convert commercially available customized DNA microarrays of any complexity and density to RNA microarrays via a T7 RNA polymerase-mediated extension of photocrosslinked methyl RNA primers and subsequent degradation of the DNA templates.

[1] Institute of Inorganic Chemistry, University of Vienna, Josef-Holaubek-Platz 2, 1090 Vienna, Austria. [2] Chair of Food Chemistry and Molecular Sensory Science, Technical University of Munich, Lise-Meitner-Straße 34, 85354 Freising, Germany. [3] Leibniz Institute for Food Systems Biology at the Technical University of Munich, Lise-Meitner-Straße 34, 85354 Freising, Germany. ✉email: mark.somoza@univie.ac.at

RNA is a key actor in many biological processes and participates in complex bioaffinity networks involving nucleic acids, proteins, and small-molecule metabolites[1,2]. These interactions are essential in all aspects of post transcriptional gene regulation from splicing to localization and translation, as well as in RNA virus infection and host defense mechanisms[3–5]. Despite its importance, the RNA interactome remains largely unexplored due to our inadequate understanding of the sequence dependence of RNA binding interactions[6,7]. Unlike DNA, which mostly exists as double-stranded structures that are well defined and exhibit low structural and binding diversity, RNA is typically single-stranded and naturally assumes complex three-dimensional structures. The combination of this conformational heterogeneity of RNA and the composite of ionic and hydrogen bonding, hydrophobic effects, π interactions, and van der Waals forces acting on binding targets, puts predictions of these interactions outside of our current reach and requires experimental approaches[8].

Massively parallel experimental assays, particularly DNA microarrays and sequencing, have revolutionized genomics research in the last decades. These same assays have also been leveraged for highly parallel quantitative investigations of the structure-function relationships of nucleic acid interactomes. For example, double-stranded DNA microarrays are used to elucidate the binding site specificities of transcription factors and other DNA-binding molecules[9–11], while affinity-based selection methods, such as SELEX, use sequencing or DNA microarrays to identify strong nucleic acid ligands after selection cycles[12–15]. These select-and-identify approaches have the advantage over on-array assays in that they can start with larger oligonucleotide pools[16,17]. However, they do not simultaneously and directly measure the affinity parameters relevant to biological function and are biased towards strongly binding sequences. In nature, low- and medium-affinity binding sites are both used by cells to enhance specificity and are relevant to therapeutic RNA-binding ligands and bioengineering applications due to the need to minimize and account for off-target effects in the presence of millions of these weaker binding sites[11,18–20].

Several techniques have been developed aiming to produce high-density RNA microarrays suitable for quantitative RNA-binding assays. The highest throughput RNA microarrays are based on capturing, sequencing, and transcribing chemically synthesized DNA libraries——with a randomized variable region to be transcribed to RNA——in repurposed Illumina sequencers[21–24]. Quantitative analysis of RNA on a massively parallel array (RNA-MaP)[21,23] starts with ssDNA sequencing libraries including an RNA polymerase promoter region, a stall sequence, and a variable region complementary to the desired RNA. After the DNA library is sequenced on the Illumina flow cell, it is enzymatically double stranded by extension of a 5′ biotinylated primer. The RNA polymerase uses this template to generate the RNA. The resulting RNA stays attached to the surface since the RNA polymerase remains stalled on the DNA template at the terminal biotin-streptavidin roadblock. High-throughput sequencing-RNA affinity profiling (HiTS-RAP)[22] is conceptually similar to RNA-MaP, but uses the E. coli replication terminator protein Tus to stall the RNA polymerase after transcribing the variable section of the DNA library. Similar sequence complexity can be achieved by combining chemical DNA synthesis and enzymatic fabrication using nucleic acid photolithography[25]. In this approach, maskless array synthesis is used to synthesize both the template DNA library 3′ to 5′ in situ on a surface and 2′-O-methyl RNA primers 5′ to 3′ using reverse synthesis. The primers, synthesized stochastically via partial photodeprotection on 50% of the surface hydroxyl groups, are then extended enzymatically using the T7 RNA polymerase

according to an adjacent DNA template. At much lower density, but with relaxed length limitations, template DNA libraries may also be spotted onto surfaces for enzymatic transcription. Including the sequence for a streptavidin-binding RNA aptamer on the template allows for the released RNA to be captured on an adjacent parallel streptavidin-coated substrate while preserving the original spatial organization[26].

The most chemically versatile approach is to use nucleic acid photolithography to directly synthesize RNA microarrays in situ using the appropriate RNA phosphoramidites[27,28]. Such direct synthesis can also accommodate much higher chemical complexity through the position-specific substitution of other nucleic acid monomers, both natural, such as DNA and 2′-O-methyl RNA, and unnatural nucleosidic and non-nucleosidic monomers[28–31]. However, direct chemical synthesis of RNA microarrays is quite inaccessible, requiring maskless photolithography equipment and supplies that are not commercially available, and the chemistry is difficult and inefficient, with an oligonucleotide length limit of roughly 30 nt[28,30].

While all these high-throughput approaches result in powerful platforms for measurements of RNA interactions, they are also highly complex and require specialized equipment and know-how beyond the practical reach of almost all researchers. In this work, we introduce a far simpler approach based on the unusual ability of the T7 RNA polymerase to synthesize RNA from a DNA template via extension of RNA or 2′-O-methyl RNA primers[25,32]. Our approach allows for DNA microarrays of any density and complexity to be converted to RNA with minimal time and effort and using only standard molecular biology equipment and supplies. We apply this method to create complex, high density RNA microarrays from Agilent SurePrint DNA microarrays as well as from DNA microarrays made via maskless photolithography.

## Results and discussion

**Enzymatic conversion of DNA to RNA microarrays.** To generate RNA microarrays, we start with conventional DNA microarrays. Spotted or synthesized in situ using ink jet printing[33], electrochemistry[34], or photochemistry[29,35], the end products are quite similar, with the 3′ termini of the DNA attached to the substrate via a short linker, and a molecular surface density of ~5 pmol/cm$^2$ due to an initial surface functionalization with silane chemistry[36]. From these similarities, we expect that DNA microarrays of almost any source can be easily converted to RNA microarrays using our approach. For conversion to RNA, the DNA sequences on these microarrays are specified to include, in addition to the template sequences, the complement to a shared 2′-O-methyl RNA primer, and a short 3′ dT homopolymer linker to the substrate (Fig. 1a). The conversion to RNA is performed in three simple steps: 1– hybridization and photocrosslinking of a 5′-psoralen 2′-O-methyl RNA primer, 2– incubation with a standard T7 RNA polymerase system resulting in primer extension according to the DNA template to yield DNA-RNA hybrids, 3– DNA template degradation using TURBO DNase. We have found that the combination of the psoralen crosslink and a short DNA dT homopolymer surface linker prevents the DNase from cleaving the linker, stably tethering the single-stranded RNA product to the surface. The details for these steps are discussed below.

To demonstrate that the approach is generally applicable to DNA microarray synthesis technologies, particularly those of the highest densities, we used an Agilent SurePrint 4 × 44 K DNA microarray (Fig. 1b). One subarray was enzymatically converted into RNA (Fig. 1b, orange framed subarray) according to the protocol shown in Fig. 1a. The fluorescent signal intensities recorded after hybridization to the RNA are spread across a fairly

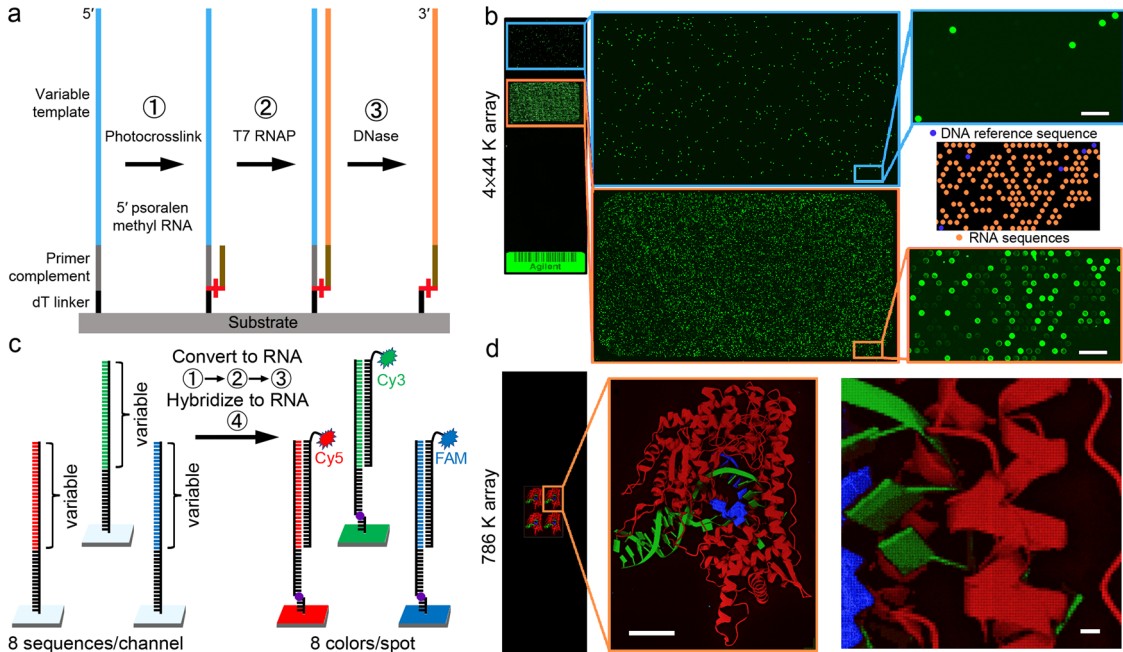

**Fig. 1 Enzymatic conversion of DNA microarrays to RNA. a** The three-step process of conversion consists of 1- primer hybridization followed by crosslinking, 2- T7 RNA polymerase-mediated RNA extension, and 3- DNase-mediated degradation of the DNA template. **b** Application of the DNA-to-RNA method to an Agilent SurePrint DNA microarray. The first subarray (outlined in blue) is untreated and hybridized with a 5′ Cy3-labeled oligonucleotide, showing the positions of sequences lacking the primer complement section. The second subarray (outlined in orange) has an identical layout but has been converted to RNA, allowing hybridization to surface-bound RNA sequences with the same Cy3-labeled oligonucleotide. Scale bars are 200 μm. In between the blue and orange-outlined subarray is a schematic illustration of the Agilent microarray layout and the corresponding location of transcribable (orange spots) and untranscribable (blue spots) DNA sequences. **c** Scheme for creating high complexity, high-resolution RGB images based on hybridization to enzymatically prepared RNA microarrays. Variable lengths of the template sequence served in realization of different shades of each color. **d** 1024 × 768 DNA array synthesized using nucleic acid photolithography such that conversion to RNA and hybridization with Cy3-, Cy5- and fluorescein-labeled oligonucleotides results in RGB images of the T7 RNA polymerase elongation complex (PDB ID 1MSW [doi.org/10.2210/pdb1MSW/pdb][40,56]). Scale bars in **d** are 1 mm (left) and 100 μm (right).

wide range that can be ascribed to variations in primer and template sequences as well as linker length (see Supplementary Fig. 1 and Supplementary Data File). DNA control sequences were included in the design and can be used to measure conversion efficiency by comparing the hybridization signal with the same fluorescently labeled probe to an untreated part of the microarray (Fig. 1b, blue framed subarray). The Agilent microarray is particularly relevant as it is a customizable and readily available product with high sequence fidelity, accommodating sequence lengths of up to 150mers[37]. Such lengths are not currently accessible through direct chemical RNA synthesis using nucleic acid photolithography[28], nor at high sequence fidelity via high density DNA photolithography[38], but if produced via enzymatic conversion from ink-jet printed DNA microarrays, are sufficient to study most RNA binding interactions, including aptamers, ribozymes, and riboswitches[39]. We also show that DNA microarrays fabricated in our lab using maskless photolithography can be enzymatically converted to RNA according to the same process (Fig. 1c, d) and produce 1024 × 768 sequences (768 K sequences) RNA microarrays. Here, an additional level of complexity has been introduced where the transcribed RNA is hybridized to a mixture of Cy3-, Cy5- and fluorescein (FAM)-labeled complements. Preliminary tests assessed the effect of the implementation of truncated DNA templates to create a range of fluorescence intensities upon hybridization (Supplementary Fig. 2). This variation in length (variable region in Fig. 1c, see Supplementary Table 1 for sequences) and the resulting range in hybridization signal was applied to generate a color gradient for each green (Cy3), red (Cy5), and blue (FAM) channel, allowing

for complex, RNA-made patterns to display colorful images (Fig. 1d). To do so, three different DNA templates with varying levels of sequence complementarity (variable region in Fig. 1c) are synthesized, transcribed into RNA, and then hybridized.

**Determination of RNA identity and process optimization.** The in situ transcription assays performed on commercial and custom-made microarrays are the fruit of preliminary tests aiming at investigating the critical parameters involved in the process of transformation of a DNA microarray into RNA (Fig. 2). The primer is a 5′-psoralen-modified 15 nt long 2′-O-methyl RNA whose sequence and length was derived from previous work[32]. Other sequence choices are expected to function equivalently, but we cannot exclude that some primer sequences and lengths are more efficient than others. The original primer choice by Daube and von Hippel was based on a DNA-RNA hybrid length of 12 nt at the 5′ end of the nascent RNA molecule within the transcription bubble of the T7 RNA polymerase elongation complex[32]. More recent structural data of a T7 RNAP complex trapped in a functional elongation mode (Fig. 1d) indicates that only a 7 bp heteroduplex is present and therefore this may also be the minimum length for primer-initiated polymerization[40]. Nevertheless, maintaining sufficient primer selectivity in the context of complex template DNA microarrays with hundreds of thousands or a few million unique sequences requires primer lengths of 12 nt or longer. The 2′-O-methyl RNA chemistry was chosen for its resistance against nuclease degradation and other RNA-like xenobiotic nucleic acids such as locked nucleic acids

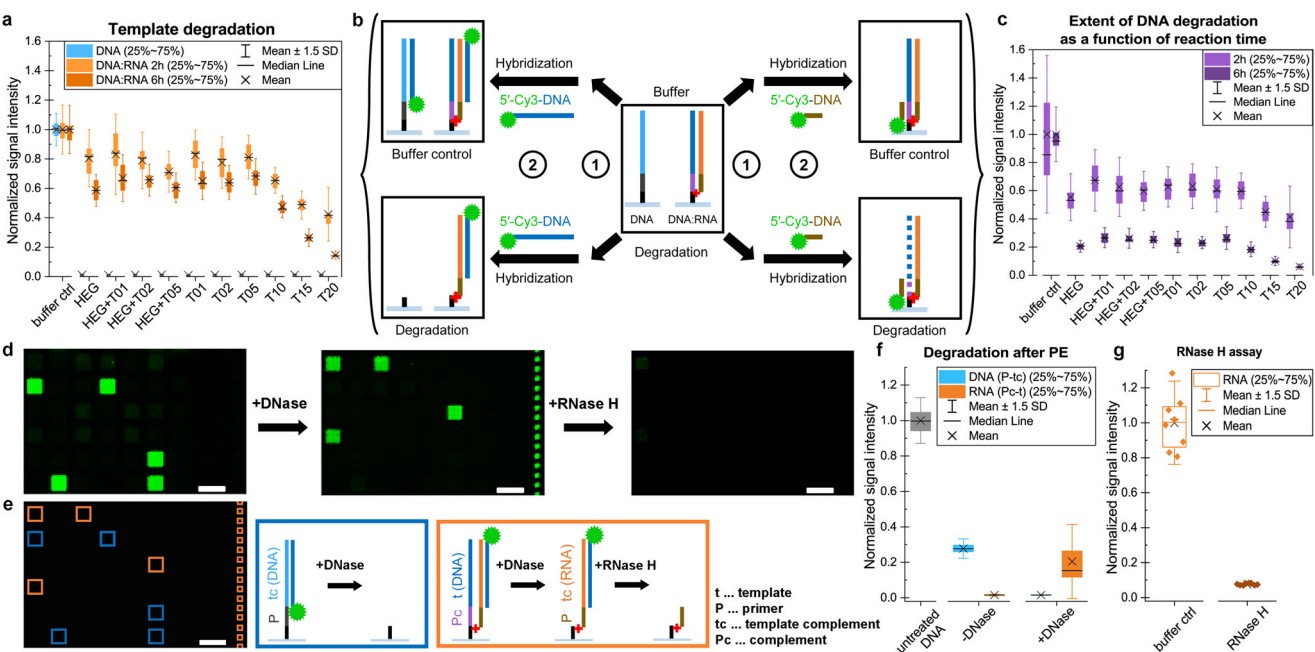

**Fig. 2 Optimization of enzymatic RNA microarray synthesis. a** DNA template degradation was assessed by mimicking successful primer extension and hybridizing either to DNA (blue) or to RNA crosslinked to template strands (orange) on different linkers after DNase treatment (24 replicates). A signal decrease for hybridization to RNA indicates degradation of the DNA linker. Fluorescence intensities are normalized to a buffer-treated control. **b** Schematic representation of the enzymatic tests performed in 2a and 2c. The microarray consists of two oligonucleotide systems: one DNA control and an RNA probe photocrosslinked to a DNA sequence. Subsections of the microarray are then either treated with buffer or degraded with DNase (step ①), followed by hybridization to a complementary strand (step ②): either to the RNA sequence (with the blue Cy3-labeled probe, assay 2a) or to the primer complement sequence (with the brown Cy3-labeled probe, assay 2c). **c** Extent of DNA template degradation as a function of treatment time with DNase for 2 h (light purple) or 6 h (dark purple) measured by hybridizing to the primer complement section of the template (24 replicates). **d** Excerpts of an RNA microarray hybridized before and after DNase and RNase H treatments. Scale bars are 100 μm. Feature identity is illustrated in (**e**) with blue squares corresponding to non-transcribed DNA template complement sequence (tc) disappearing after DNase treatment and orange squares populated with transcribed DNA template sequences (t), resulting in RNA product with the template complement (tc) sequence, available for hybridization after template degradation. The resulting DNA-RNA hybrid can then be recognized and processed by RNase H, leading to RNA degradation and the corresponding RNA features disappearing from the microarray scan. **f** Signal intensities recorded for hybridization to DNA or RNA at various stages of the enzymatic primer extension (PE) process, the blue bars correspond to non-transcribed DNA (blue features in (**e**)) and orange bars to RNA (orange features in (**e**)) (62 replicates). PE efficiency can be measured relative to the intensity of hybridization to ssDNA primer-template complement (P-tc) sequences on an untreated DNA microarray (gray bar). **g** Signal intensities recorded for hybridization to ssRNA without and with RNase H treatment (8 replicates). Source data are provided as a Source Data file.

(LNA)[41] or 2′-deoxy-2′-fluoro-RNA (2′-F-RNA)[42] may provide similar protection while increasing selectivity. The photo-crosslinking approach has been described in detail elsewhere[29] and the ideal conditions were found to be the use of 5′-psoralen-labeled oligonucleotides forming a TA base-pair adjacent to the psoralen moiety for proper intercalation and photocrosslinking, which is performed under 365 nm UV light at ~4 °C (see "Methods" section). We then focused on the degradation of the DNA template without cleaving the linker to the surface, a pivotal aspect of the method as it is necessary to retain the polymerized RNA immobilized to the surface.

The use of deoxyuridine (dU) nucleotides in the template sequence, followed by incubation with uracil DNA glycosylase (UDG) after conversion to RNA is effective and unproblematic in the context of our own microarray synthesis[43], but dU-containing microarrays are not available commercially, limiting the general usefulness of this strategy. We also experimented with template degradation with the *E. coli* lambda exonuclease. This exonuclease removes nucleotides 5′ to 3′ in single- and double-stranded DNA[44], and we speculated that it might also work in DNA-RNA hybrids and would be unable to bypass the crosslink. However, we were unable to observe DNA template degradation using this exonuclease. Finally, we were successful with TURBO DNase, an optimized variant of the DNase I endonuclease. This

endonuclease is less active on DNA-RNA hybrids (vs dsDNA), but is even less active on ssDNA, and preferentially cleaves at purine-pyrimidine junctions, favoring the integrity of the ssDNA homopolymer linker[45]. We also speculated that steric interference by the glass surface would prevent endonuclease activity in the case of sufficiently short linkers. To verify this, we considered various linker types and linker lengths, from homopolymeric dT (1–20 nt in length) to non-nucleosidic hexaethylene glycol (HEG), and treated both DNA and RNA with TURBO DNase in a system mimicking successful enzymatic extension of the primer (Fig. 2b and Supplementary Fig. 3). We measured the extent of degradation by hybridization to ssDNA and to immobilized RNA following either enzymatic treatment or incubation in buffer (Fig. 2a and Supplementary Fig. 3). The ssDNA is thoroughly degraded in all cases after 6 h while hybridization to the RNA remains possible except for dT linkers ≥15 nt where signal strength starts to decrease (loss of ~70% of fluorescence intensity), indicating that the linker itself becomes a substrate for the DNase enzyme. To minimize cleavage, we thus settled on a dT$_5$ linker.

Acknowledging the lower efficiency of TURBO DNase when acting on DNA-RNA hybrids, we became interested in studying how far the enzyme had processed the DNA substrate and how thoroughly the primer sequence of the DNA template was

degraded (Fig. 2c and Supplementary Fig. 3). Indeed, the signal corresponding to base-pairing to the DNA section complementary to the primer decreases by only 40–50% upon 2 h treatment. A treatment of 6 h results in significantly improved degradation of the DNA template, with only up to 25% residual fluorescent intensity compared to incubation in buffer for the same period of time. Template degradation is a prerequisite for detection of polymerized RNA through hybridization to a fluorescently labeled complementary probe (Fig. 2d), which naturally corresponds to that of the DNA template (Fig. 2e). Hybridization with a fluorescent probe is inefficient before template degradation presumably due to the difficulty in displacing the template strand from the DNA-RNA heteroduplex. After DNase-mediated degradation, the RNA is single-stranded and fully available for hybridization and the RNA features on the microarray become visible upon fluorescence scanning (Fig. 2d, f). The same scan shows the disappearance of the hybridization signal to the untranscribable DNA, suggesting correct degradation. With the template degradation complete and the linker intact, we compared hybridization intensities between ssRNA and an ssDNA of the same sequence (which was not subjected to the enzymatic synthesis process) as a proxy to evaluate transcription efficiency (Fig. 2f). We found that the efficiency of the process is between 15% on commercial microarrays (Supplementary Fig. 1b) and ~20% on photolithography microarrays (Fig. 2f). Finally, an RNase H assay was carried out to verify the identity of the product oligonucleotide. As expected, a short enzymatic treatment of the RNA:DNA duplex resulted in a complete loss of fluorescence (Fig. 2d) and the inability to rehybridize (Fig. 2g), indicating full degradation of an RNA strand.

**RNA product analysis.** To determine if full-length RNA is efficiently produced by the T7 RNA polymerase by extension of 2′-*O*-methyl RNA primers crosslinked to ssDNA templates, we replicated the array conversion in solution using commercially synthesized oligonucleotides: a DNA template, a 5′-psoralen 2′-*O*-methyl RNA primer, and a chimeric sequence consisting of the primer and the full-length RNA product. The first two of these were photocrosslinked and either digested directly with DNase or first used to synthesize RNA and then digested. Polyacrylamide gel electrophoresis (PAGE) analysis (Fig. 3a) demonstrates that the T7 RNA polymerase generates the full-length RNA product. Shortmers in lane 3 may be larger DNA template degradation fragments also seen in lane 2.

The production of full-length RNA was further confirmed with on-array experiments using fluorescently-labeled UTP. DNA templates containing single dA nucleotides at every third position were converted into RNA using a standard nucleoside triphosphate (NTP) mix supplemented with fluorescent Cy3-UTP, yielding fluorescent RNA products (Fig. 3b). After an initial decrease in intensity, the fluorescence from the Cy3-rU as it is incorporated closer to the 3′ end of the RNA remains constant within the final 20 nucleotides, at about 70% relative to the first position, indicating predominant synthesis of the full-length product. The decrease of fluorescent signal intensities with increasing distance of the fluorophore from the glass surface is consistent with previous observations upon hybridization of Cy3-labeled oligonucleotides to surface-bound DNA[46] (compare also Supplementary Fig. 4). This suggests that at least some of the observed decrease in fluorescence is an artifact of the increase of oligonucleotide length rather than reduced yield of full-length RNA.

In summary, we have shown that: (i) the T7 RNA polymerase efficiently extends 2′-*O*-methyl RNA primers photocrosslinked to ssDNA template arrays, resulting in DNA–RNA hybrids; (ii)

TURBO DNase degradation of the DNA template results in RNA bound to the original position of the template via the 2′-*O*-methyl RNA primer crosslinked to the original short dT linker; and (iii) the combination of primer photocrosslinking and DNase degradation results in a universal strategy that enables the conversion of standard commercially-available DNA microarrays of any density and complexity to complementary RNA. Thus, functional high-density RNA microarrays can be generated overnight with minimal effort and using only equipment, supplies and skills commonly found in molecular biology research environments. Although RNA is likely to remain the most useful product, the approach can be extended through the use of engineered polymerases[47] to other primer chemistries such as DNA, RNA, and LNA, but also to the production of complex high-density arrays of stably-linked dsDNA, 2′-fluoro- and 2′-azido-substituted analogs for aptamer and siRNA applications[48], as well as to the conversion of RNA with 5-methylcytosine and pseudouridine substitutions for research in modified mRNA therapies[49].

## Methods

**Sequence design.** DNA sequences intended for conversion to RNA consist of a 3′ dT linker to the surface, a segment complementary to the primer, and a variable segment that serves as a template for transcription. The optimal dT linker length for our own microarrays synthesized via photolithography was identified being 5 to 10 nt, whereas the optimal length for the Agilent microarrays was found to be 10 nt (see Supplementary Fig. 1). Our experiments using hexaethylene glycol as non-nucleosidic linker do not show significantly higher yield of RNA compared to DNA linkers (see Supplementary Fig. 4b). In order to simplify the synthesis procedure, we, therefore, suggest the use of dT linkers. Microarray control sequences without the primer complement do not function as templates for the polymerase. With their sequence being complementary to the template, they were used to verify microarray synthesis quality and as intensity references by omitting DNase degradation and by hybridizing with the Cy3-labeled complementary sequence.

**Sequence design for RGB image.** In order to generate a visually compelling image demonstrating the ability of our approach to convert DNA microarrays to RNA microarrays at high spatial resolution and without spatial artifacts, we selected a relevant color image showing the structure of the T7 RNA polymerase elongation complex (PDB ID 1MSW [https://doi.org/10.2210/pdb1MSW/pdb])[40] bound to the DNA template and RNA product. It is a shift from the initiation complex to this elongation complex, even in the absence of promoter initiation that appears to enable the elongation of RNA primers by this polymerase[32]. A 256 × 192 pixel color representation of the crystal structure was converted in Photoshop to an RGB bitmap with 8 brightness levels per color channel. Four such images were used to tile the available synthesis space of 1024 × 768 pixels. Each of the color channels is represented by one template sequence: TCACCGAATCGATTCCATCTGCTTC (red), TCAACCCAGGTCCAATTTCC (green), and ACAGTGGATCGTACTCA GGTCTCA (blue). Intensity levels are achieved by 5′ truncations to these sequences (Supplementary Table 1). Initial hybridization experiments with the RNA conversion product of all possible truncations of these sequences (Supplementary Fig. 2) allowed a selection of 8 of these spanning the full intensity range with uniform intensity spacing (Supplementary Table 1). The resulting intensity range allowed for realization of different shades of each individual color conveyed by the fluorophores Cy5, Cy3, or fluorescein (FAM) labeled oligonucleotides complementary to the product RNA, for the generation of an RGB image with eight different levels of each color (Supplementary Fig. 5). To enable conversion to RNA, all template sequences shared a common TTCGCCGTGTCCCTATTTTT sequence at the 3′ end serving as linker and primer hybridization/crosslinking site.

**Agilent microarray.** A 4 × 44 K SurePrint custom DNA microarray (AMADID 086693) was designed following the aforementioned criteria. Design characteristics for the Agilent microarray are provided in Supplementary Table 2. The design includes variations in linker length (dT*n*, *n* = 5, 10, 15, 20), permutation of the three 3′ terminal positions of the sequence complementary to the primer, and permutations of three central positions of the template sequence to generate mismatches. These changes vary the efficiency of transcription and, after transcription, hybridization efficiency between the RNA and a Cy3-labeled complementary DNA oligonucleotide. The Agilent microarray design also includes DNA sequences complementary to this fluorescent oligonucleotide, allowing direct hybridization as a reference for hybridization signal intensities for the detection of RNA product as shown in Fig. 1b. In total, the microarray design contains 60 replicates of each sequence. Custom-made hybridization chambers (Grace Bio-Labs SecureSeal RD500958), similar to those used in the case of the microarrays we synthesize ourselves using photolithography, allowed for applying and exchanging

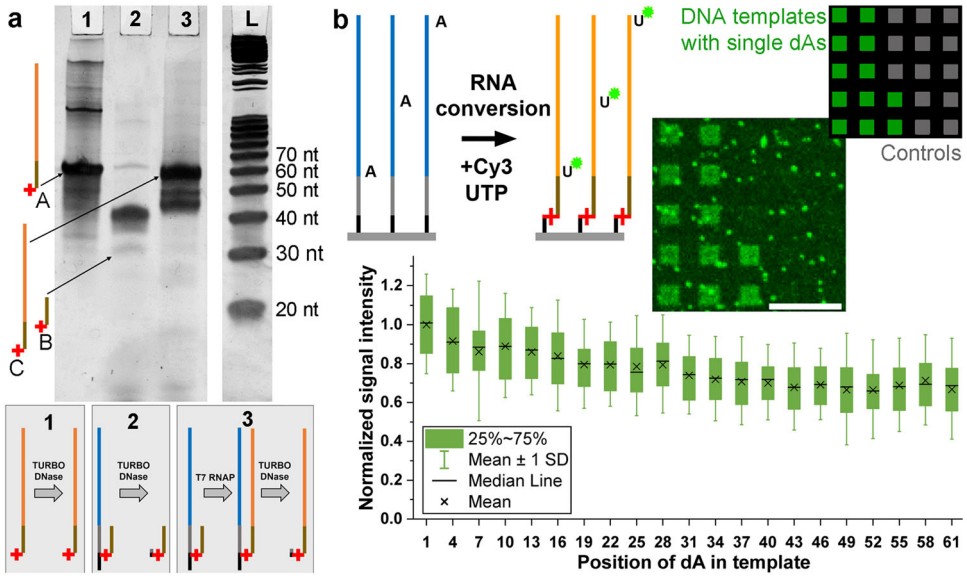

**Fig. 3 Efficiency of 2′-O-methyl RNA primer extension by the T7 RNA polymerase. a** PAGE analysis of primer extension in solution. Lane 1: a 57 nt reference product oligonucleotide made using conventional solid-phase synthesis and consisting of a 5′-psoralen 27 nt 2′-O-methyl RNA section and a 3′ 30 nt RNA section (arrow A) plus TURBO DNase. Lane 2: 62 nt DNA reference of template and linker made using conventional solid-phase synthesis and crosslinked to the 27 nt 2′-O-methyl RNA primer, followed by DNase degradation. Degradation products are the primer crosslinked to DNA fragments and residual primer (arrow B). Lane 3: product of the full enzymatic conversion and template degradation of the conventionally synthesized and crosslinked oligonucleotides used in lane 2, with arrow C showing the full-length RNA extension of the 2′-O-methyl RNA primer. The steps performed on each sample loaded into lanes 1-3 are illustrated in the scheme below. **b** The efficiency of on-array primer extension of in situ synthesized DNA is demonstrated using templates with a single dA introduced at every third position within non-dA mixed-base 61 nt sequences. Conversion is indicated by fluorescence of incorporated Cy3-UTP in the RNA strands (85 replicates). Scale bar is 100 μm. Source data are provided as a Source Data file.

the reaction mixtures on individual Agilent subarrays while keeping the other subarrays dry. One subarray was subject to the workflow described below for hybridization, photocrosslinking, enzymatic primer extension, and template degradation. To assess the conversion to RNA, hybridization with t-Cy3 (5′-Cy3-TCAACCCAGGTCCAATTTCC, IDT) was performed in parallel on this subarray and an adjacent subarray on the same slide. To align the scan with the sequence layout information provided by the manufacturer, GenePix 6.1 (Molecular Devices) software was used followed by further data analysis in Microsoft Excel.

**Hybridization and photocrosslinking**. DNA template arrays where hybridized with 89 nM primer oligonucleotide (2′-O-methyl RNA, 5′-Ps-UAGGGA-CACGGCGAA, Eurogentec) in MES 1× buffer (100 mM MES, 1 M Na$^+$, 20 mM EDTA, 0.01 % Tween-20) supplemented with 0.44 mg/mL acetylated BSA (Pro-mega R3961) in self-adhesive chambers (Grace Bio-Labs) for 30 min at 37 °C with rotation (rotating bubble-based mixing), followed by washes with: 1– non-stringent wash buffer (NSWB) (6× SSPE, 0.01% Tween 20) for 1 min, 2– stringent wash buffer (SWB) (100 mM MES, 0.1 M Na$^+$, 0.01% Tween 20) for 30 s and 3– 5 s in final wash buffer (FWB) (0.1× SSC). The primer was then crosslinked to the DNA template at 4 °C in 1× MES buffer by exposure to 25 J/cm² of 365 nm UV light. Crosslinking was performed using a custom-built light source based on the same high-power Nichia NVSU333A 365 nm UV LED used for the DNA template microarray photolithographic synthesis. The radiant power of the crosslinking light was determined using SÜSS Model 1000 UV intensity meter with a 365 nm probe. A more detailed description of the light source and the setup for photocrosslinking is given in Supplementary Fig. 6. Unlinked primer was removed by pipetting RNase-free water in and out of the hybridization chamber for one minute. The crosslinking protocols as well as extensive details on the efficiency of on-array photocrosslinking reactions on microarrays have been previously published[29].

**Primer extension and template degradation**. Next, a solution of New England BioLabs (NEB) 1× RNA Pol Reaction Buffer with 0.2 mg/mL acetylated BSA and 0.005% Triton X-100 (Sigma-Aldrich) was applied to the hybridization chamber, followed by incubation for 30 min at 37 °C with rotation. After removal of this solution, a mix of 3 units/μL T7 RNA polymerase (NEB M0251) in 1× RNAPol reaction buffer supplemented with 0.005% Triton X-100 (Sigma-Aldrich), 0.5 mM of each NTP (Thermo Scientific R0481), 1 unit/μL RNase inhibitor (NEB M0307) and 4 mM DTT (Sigma-Aldrich D5545) was applied to the array. For incorporation of a fluorescent label into the RNA product, Cy3-UTP (Jena Bioscience NU-821-CY3) was added at a final concentration of 0.0175 mM and UTP at 0.2 mM. The enzymatic primer extension reaction was stopped after 16 h at 37 °C with rotation by exchange of the solution with a 0.1 units/μL mix of TURBO DNase

(Invitrogen) in 1× buffer supplemented with 1 unit/μL RNase inhibitor for degradation of the DNA template (5 h at 37 °C with rotation). The degradation reaction mix was pipetted out and discarded, and the respective array briefly washed with NSWB by pipetting in and out two volumes of buffer within the hybridization chamber. After removal of the hybridization chamber, the entire slide was washed briefly in FWB and dried with a microarray centrifuge.

**Hybridization-based detection**. Hybridization with a fluorescently-labeled oligonucleotide probe complementary to the RNA generated was carried out for RNA detection. The hybridization mix consisted of 12.2 nM probes (see Supplementary Table 3 for sequences) in MES 1× buffer supplemented with acetylated BSA (0.44 mg/mL) and 0.07 units/μL RNase inhibitor, and incubation was performed for 1 h at 37 °C with rotation. After hybridization, the microarray was washed for 2 min in NSWB, 1 min in SWB, and 10 s in FWB. In the case of fluorescein (FAM)-labeled probes, the slide was additionally washed for 30 s in carbonate/bicarbonate buffer (pH 10) to enhance fluorescence[50]. After drying in a microarray centrifuge, microarrays were scanned with a GenePix scanner (Molecular Devices) at 2.5 or 5 μm resolution. Fluorescent signal intensities were extracted using NimbleScan 2.1.68 (NimbleGen), followed by data analysis in Microsoft Excel. For maximum comparability between effects of differing reaction conditions, the layout of the microarray was adjusted to divide the synthesis area in four parts with identical sequences. Custom-made self-adhesive chambers (Grace Bio-Labs RD475732-M) allow to individually address each subarray. The efficiency of the conversion procedure was determined by parallel hybridization to control sequences of DNA in an untreated and RNA as product in a treated subarray on the same surface.

**RNase H assay**. The RNase H degradation assay was performed in order to confirm the identity of the conversion product. Following detection of the microarray-bound RNA via hybridization with Cy3-labeled complementary DNA, the RNA microarray was treated with 0.02 units/μL RNase H (stock: 5 units/μL, NEB M0297) in 1× buffer (50 mM Tris-HCl, 75 mM KCl, 3 mM MgCl₂, 10 mM DTT, pH 8.3 at 25 °C) in an adhesive hybridization chamber for 1 h at 37 °C with rotation. As a control, another sub-array with RNA on the same slide was treated in parallel with 1× buffer. After discarding the reaction mix, washing was performed in the same way as after DNase degradation. In order to assess the completeness of the degradation, the initial hybridization-based detection with Cy3-labeled complementary DNA was repeated, and the original presence of RNA was confirmed by a near complete loss in hybridization signal following RNase H treatment (see Fig. 2d and Supplementary Fig. 4c, d).

**Cy3-UTP-based detection**. Template sequences consisted of the dT linker and primer complement followed by 20 concatenations of TGC with a single dA at the beginning or at each of the 20 possible positions following a dC (see Supplementary Table 4). This particular pattern was selected in order to ensure similar sequence context of Cy3 in the RNA product strand, as the impact of the nucleobases in close vicinity to the fluorophore on its signal intensity has been reported[51]. To account for the background signal due to non-specific Cy3-UTP binding to the glass surface (Fig. 3b), control sequences without the primer complement were also included. In order to further account for spots of background fluorescence, signal intensities of all 85 replicates of each template and control sequence on the array are averaged. For data analysis, averaged signal intensities from control sequences are subtracted from those detected for template sequences of corresponding type regarding the position of dA. These corrected values are normalized to the highest corrected signal.

**Polyacrylamide gel electrophoresis analysis**. In order to emulate the array conditions, 50 pmol of a DNA template strand (Eurogentec) consisting of a 3′ dT 5mer followed by a sequence complementary to the primer, followed by a 30mer sequence serving as template for transcription was annealed with an equal amount of the 5′-Ps-2′-O-methyl RNA primer (Eurogentec) (see Supplementary Table 5 for sequences) in 1.5× transcription buffer (Invitrogen; 200 mM Tris-HCl pH 7.9 at 25 °C, 30 mM MgCl$_2$, 50 mM DTT, 50 mM NaCl and 10 mM spermidine) for 5 min at 70 °C, followed by a gradual cooling to room temperature for 1.5 h. Crosslinking was performed at 4 °C and 50 J/cm² 365 nm light (setup shown in Supplementary Fig. 6c). The primer extension reaction consisted of 2 mM of each NTP, 2 units/µL RNase inhibitor, and 3 units/µL T7 RNA polymerase. The reaction was incubated at 37 °C for 16 h. The DNA template was degraded for 1.5 h at 37 °C after the addition of 1 unit/µL TURBO DNase in 1× TURBO DNase buffer to reach a final concentration of 0.1 units/µL (Fig. 3a, lane 3). In order to demonstrate the effect of primer extension, another sample was prepared in exactly the same way, but skipping the addition of RNA polymerase to the primer extension reaction mix (Fig. 3a, lane 2). To ensure similar buffer conditions in order to avoid differences in running speed between samples and control, 50 pmol of a chemically synthesized "product" probe (Eurogentec)—which consists of a 5′-Ps-2′-O-methyl RNA primer section and an RNA part, corresponding exactly to the product aimed for in the reaction—were diluted in 1.5× transcription buffer, supplemented with the reagents used in primer extension except for RNA polymerase being replaced by the corresponding volume of water (Fig. 3a, lane 1). Non-transcribed controls were also subjected to DNA degradation with TURBO DNase as before. The length standards, Ladder 20/100 (IDT 51-05-15-02) and 100 bp DNA ladder (NEB N3231), were used as 1:1 mixture and diluted in the primer extension solution, omitting the RNA polymerase. All samples were then mixed with 2× RNA loading dye (NEB B0363), denatured for 10 min at 70 °C, and placed on ice immediately. After briefly spinning down, they were loaded and run on a 12% denaturing polyacrylamide gel (7 M urea; 19:1 ratio acrylamide:bis-acrylamide) with 0.5× TBE as running buffer. Finally, the gel was stained for 30 min in a 1× solution of SYBR Gold (Invitrogen S11494) in 1× TBE buffer and scanned with a Fusion FX7 imaging system (Vilber Lourmat).

**Degradation with DNase**. DNA template degradation was evaluated as an isolated processing step by hybridization and photocrosslinking of a chemically synthesized "product" probe, mimicking the product of enzymatic primer extension by displaying the structure of 5′-Ps-2′-O-methyl RNA with an additional RNA section at the 3′ end (see Supplementary Table 5 for sequence), to surface-bound DNA according to the aforementioned protocol. Two versions of DNA sequences were synthesized on the microarray: one version that potentially serves as a template for conversion to RNA (therefore allowing for hybridization and crosslinking of the "product" probe in this particular experimental setting), the other version with a sequence that is identical to the RNA section of said "product" probe. Control and template sequences were synthesized on linkers of different lengths by combination of hexaethylene glycol and/ or dT units, resulting in total in 24–25 replicates of each sequence variant. Following the processing steps for "hybridization-based detection", the reference signal for hybridization to crosslinked RNA and ssDNA was determined. A microarray layout with four identical, individually addressable sections was used. Individual sections were subject to degradation for 2 or 6 h in a 0.1 units/µL mix of TURBO DNase (Invitrogen) in 1× buffer supplemented with 1 unit/µL RNase inhibitor, or incubated for 2 or 6 h in 1× buffer supplemented with 1 unit/µL RNase inhibitor ("buffer control") at 37 °C with rotation. The solutions were pipetted out of the reaction chambers and the slide was washed with NSWB and FWB. The signal intensities for hybridization to ssDNA and crosslinked RNA were measured and compared between the sections subject to the enzyme treatment and the control in order to assess the effect of TURBO DNase on the presence of surface-bound DNA and RNA. Detection of residual DNA in duplex with RNA was performed in another hybridization experiment, using a fluorescently labeled probe with the nucleotide sequence of the 2′-O-methyl RNA primer section (Fig. 2a, c).

**Photolithographic synthesis of template DNA arrays**. We use a custom-built system consisting of an optical imaging system that projects the mirror pattern of an XGA (1024 × 786) digital micromirror device (Texas Instruments) onto a functionalized glass slide in a reaction chamber[52]. The DMD is illuminated with

light from a 365 nm LED (Nichia NVSU333A)[53]. An Expedite 8909 nucleic acid synthesizer is coupled to the reaction chamber and pumps reagents and solvents in synchrony with the UV light exposures. The DNA synthesis cycle is a variant of the standard phosphoramidite chemistry, customized with phosphoramidites (Orgentis) with the photolabile BzNPPOC group on the 5′ hydroxyl[54,55]. Monomer coupling times were set to 15 s for BzNPPOC-dN, 300 s for NPPOC-protected hexaethylene glycol phosphoramidites (ChemGenes), and 120 s for DMTr-dT phosphoramidites. Photodeprotection exposure was 3 J/cm² or 6 J/cm² to remove BzNPPOC and NPPOC protecting groups, respectively. The following solvents and reagents used in synthesis were purchased from Sigma-Aldrich: acetonitrile (34851), DCI activator (L032000), both kept dry under molecular sieves (Z509027), exposure solvent composed of 1% (m/v) imidazole (56750) in DMSO (34943), and 20 mM iodine in H$_2$O/pyridine/THF (L060060) as oxidizer. TRIDIA Activated Slides (Surmodics) were used as substrates. After synthesis, all protecting groups were removed in 2 h in a 1:1 (v/v) ethylenediamine/ethanol solution in a staining jar.

**Statistics and reproducibility**. All analyses were performed in OriginPro 2020 (version 9.7.0.188) and Microsoft Excel 2020. No statistical methods were used to determine sample size and no statistical tests were performed on the data. No data were excluded from the analyses, except for two data points (out of 85) in the x-axis category "7" (concerning the position of dA the template) in Fig. 3b due to a piece of dust on the surface at these positions causing extraordinarily high fluorescent signal. The experiments were not blinded, nor randomized, except that the positions of microarray probes were randomized on the surface when applicable. The microarray experiments in Fig. 1 were performed a single time each. Both the RNase degradation assays, as shown in Fig. 2d (step 2) and Supplementary Fig. 3d, and the degradation assays using TURBO DNase (scan shown in Fig. 2d, step 1) were performed four times. PAGE analysis of primer extension in solution was carried out according to the procedure described herein independently twice to yield results as shown in Fig. 3a.

**Reporting summary**. Further information on research design is available in the Nature Research Reporting Summary linked to this article.

## Data availability

Raw image files of microarray scans and the design file of the Agilent microarray (AMADID number 086693) are available at https://doi.org/10.5281/zenodo.6636404. The crystal structure of the T7 RNA polymerase elongation complex has the PDB ID 1MSW [https://doi.org/10.2210/pdb1MSW/pdb]. Source data are provided with this paper.

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

## Acknowledgements

This research was funded in part by Austrian Science Fund (FWF) grants P23797, P27275, and P30596 to M.S and P34284 and I4923 to J.L. Additional support by the University of Vienna and its Faculty of Chemistry is gratefully acknowledged.

## Author contributions

M.S. conceived of the method. E.S. performed the experiments based on initial results obtained by E.S., K.H., and J.L. All authors undertook the data analysis and discussed the results. E.S, J.L., and M.S. wrote the final manuscript.

## Funding

## Competing interests

The authors declare no competing interests.
