## [Peer Review File · Nature Communications]

REVIEWER COMMENTS

Reviewer #1 (Remarks to the Author):

While this is an interesting topic and interesting work, and the technical quality of the study seems fine, the paper is severely remiss in one important regard.

The basic idea presented in this paper has been published previously in 2014 by another group in the authors cited reference 15. Reference 15 is cited in a somewhat inscrutable manner in the introduction as follows: "Similar sequence complexity can be achieved by combining chemical DNA synthesis and enzymatic fabrication using nucleic acid photolithography¹⁵, or also enzymatically, but at lower density, via capture of RNA from DNA template arrays¹⁶."

In fact the procedure using in reference 15 is conceptually identical to that used in the present paper, with the elements of a photolithographically fabricated DNA array being copied into RNA by T7 DNA polymerase, followed by enzymatic destruction of the DNA to leave only the RNA on the array. This fact was not recognized in the paper. The differences are in the details of the chemistry employed; it may be that the authors implementation is more practical and useful to the world, so I am not saying the result is without value - but it is not conceptually new, and probably does not belong in Nature Communications for that reason. I think the authors need to rewrite the paper to correctly position it relative to the work in reference 15 and then submit to a somewhat less prestigious venue with a transparent acknowledgement of its follow-on nature but some reasonable motivation for the work in its practical utility to the community.

Reviewer #2 (Remarks to the Author):

Summary:

Consistent with the title, this paper essentially describes a means to convert single-stranded oligonucleotides, often synthesized on arrays in which the DNA is coupled to the surface at its 3' end, into an equivalent array of RNA. This is done by first hybridizing a primer containing the T7 polymerase promoter sequence. The primer contains 5'-psoralen 2'-O-methyl, which can be crosslinked to the DNA

strand. The T7 polymerase then transcribes the remainder of the oligonucleotide, then the DNA strand is degraded by TURBO DNase, leaving only single-stranded RNA with the complimentary sequence.

The technique is of obvious utility and would be of general interest as it represents a new method in general molecular biology. It may in fact be transformative for specialists who study RNA binding proteins, RNA folding, etc.

The manuscript itself would benefit from a great deal of revision, so that others can actually use the technique and understand how it works.

The main text reads reasonably well, but that is because it is something of an advertisement. Exploration of many critical experimental parameters is relegated to the Supplementary document, and there is almost no connection between the main text and the Supplementary. The Supplementary Figures and Tables are not referenced or discussed in the main text, and there is little to no integration between the two. In addition, the Supplementary is very difficult to follow. There is no obvious overall organization, there are missing axis labels and no description of samples shown, etc. It also contains “data not shown” numerous times, which is inappropriate given that there are typically no space constraints on the Supplementary document.

As a consequence, the uninitiated reader (even a supposed expert in the field) is left with a great deal of decoding and inference. Without better presentation and integration of the Supplementary, it is hard to know what is scientifically most important. As this paper is about a method, it really must describe how the method really works, how it was developed, and how the critical parameters were explored. From the length of the Supplementary, I infer that that there has in fact been considerable exploration of parameter space.

I have enumerated a number of specifics below, which may not be comprehensive. I would suggest that much of the Supplementary should be moved to the main text. As an example, the length and sequence of the oligo-dT linker appears to be a critical parameter, and indeed it is surprising that it is not cut by the DNase – presumably due to steric incompatibility with the array surface? It is claimed that its length was optimized, and I can see that there are some kinds of experiments in the Supplementary, but I cannot decipher how they constitute optimization. There must be a tradeoff between efficiency of extension and prevention of DNase cleavage.

Note that I am not proposing additional experiments. Only that what has been done needs to be presented in a more sensible and easily digested fashion. Questions below are simply what came to mind and if the answer is known it should be included.

Also, I should mention that these are running notes – I struggled to prioritize, and instead I am offering these comments as a starting point for overall reorganization of the paper.

Introduction:

The first paragraph is devoid of references and is overly general – there is no need to recite common knowledge. The comparison between DNA and RNA doesn't seem relevant and, the way it is presented, seems to suggest that DNA-target interactions don't generally involve hydrogen bonding, van der Waals forces, etc. which I don't think was the intent.

For the 2nd sentence, “These interactions are essential in all aspects of gene expression from splicing and post-transcriptional regulation to localization and translation ...”, could be more accurately phrased as, “... essential in all aspects of post transcriptional gene regulation from splicing to localization and translation ...”

2nd Paragraph: Citations 4 and 5 seem incorrect. Both studies analyzed RNA, primarily. However, the paper that DNA is being analyzed (e.g. “DNA pull-down arrays”, etc.). And, as far as I recall, neither citation used SELEX.

3rd Paragraph: Please list specific techniques being referenced (e.g. HiTS-RAP, RNA-MaP, etc.). The 2nd sentence which attempts to summarize references 11-14 is unclear. Also, terms such as “highest throughput” or “ultra-high-throughput” (and “very high throughput” — in the Abstract) are not meaningful.

Also, one potentially confusing aspect is use of the term “array”, which is commonly used in the literature to refer to microarrays. However, some of the high-throughput sequencing platforms cited in the text are being referred to as “arrays”. This is technically correct but nevertheless can create ambiguity. To avoid confusion, I would suggest using the full name for microarray (ie avoid abbreviating microarray to “array”) and/or terms such as “chip arrays” (for microarrays), “flow cell arrays” (for the high-throughput sequencing based systems), etc.

Results:

The two sections, “Enzymatic conversion from DNA arrays” and “Conversion to RNA of a commercial DNA array”, could easily be consolidated into one (e.g. “Conversion of DNA to RNA microarrays”).

There is very little text corresponding to Figures 1b, 1c or 1d. Specifically, there is one sentence saying an Agilent and a photolithographic were converted from DNA to RNA arrays. Which is followed by an entire paragraph on customizable Agilent arrays which seems better suited for the Introduction. Also, what is the relevance of these figures? Integration with the Supplementary Materials would be of benefit here. Importantly, it is not clear what the authors are trying to present in Figure 1d and Supplementary Figure 5.

The Figure Legend for Fig 1 reads, “Enzymatic conversion of DNA arrays to RNA. (a) Three-steps for converting an existing DNA array to RNA. (b) Conversion of an Agilent SurePrint DNA microarray to RNA.” — clearly, the text is repetitive and could be improved. The sentence, “The second array has an identical layout but has been converted to RNA to allow hybridization to surface bound RNA sequences” seems incorrect. Should it say something like, “to allow hybridization to Cy3-labeled oligo, as in the first sub-array”? Also, the 1st and 2nd sub-arrays are boxed in blue/red — this should be mentioned in the Figure Legend.

In Figure 1b, there is a microarray image with red and blue spots. It is unclear what this is — I assume it is a false-colour overlay, but I cannot find a description in the Figure Legend or main text.

For Figure 1b, the spot intensities in the converted Agilent RNA array have substantial variability. The distribution of the median fluorescent spot intensities across the Agilent array should be shown. For the Cy3-UTP incorporation in panel 2b (photolithographic array), dA template positions greater than 40 have decreased fluorescent intensities which indicates a reduction in full-length RNA. This is paralleled by the smaller products (~43-50 nts) derived in solution in Figure 2a — presumably, these are RNA (?). How can these issues be explained and remedied? Perhaps the DNA to RNA conversion is not efficient as expected, or RNA detection using the Cy3-labeled oligo hybridization is suboptimal? On-array RNA conversion of DNA using Cy3-UTP (in Agilent arrays) could help address this issue. Is it possible that the DNA template sequences are truncated or is the T7 RNA polymerase prone to prematurely terminating from certain templates?

Again, I do not know what is the point of Figure 1d, it does not appear to show any meaningful data or information. The structures are from PDB. It is unusual to re-present data from another paper without appropriate context.

For Figure 2a, it would be helpful to have illustrations corresponding to the products analyzed by PAGE. For Lane 3, there are 2 two prominent smaller bands. Are these RNA or products from DNA digestion? This issue could be addressed via RNase digestion.

For Figure 2b, there is significant background fluorescence spots throughout the microarray. Of course, this would impact quantification and a downstream analysis. How is this issue addressed?

Methods:

There are two different “Methods” sections. I assume this was an accidental insertion of the second “Methods” header.

The methods for microarray “Sequence design” only discuss dT length. No other information about the Agilent microarray sequences is given. An AMADID number is not provided as far as I am aware. Additionally, four different fluorescently labeled probes were used — all of which have different lengths and sequences. It is not clear how these sequences were selected or where they bind.

More information regarding bias and/or efficiency during RNA primer extension via T7 RNA polymerase as well as comparison between RNA primer- and T7 promoter-mediated RNA transcription would be useful information to include in the manuscript.

As the dT and RNA linker remain after chemical/enzymatic treatment of microrarrays, what specific optimizations could researchers use to prevent background and/or undesirable interactions (e.g. during RNA-RNA or RNA-protein interaction experiments, for example) in these regions?

What rules govern sequence selection for the RNA linker? Are there considerations for length and/or sequence that the readers should be aware of? For example, the RNA linker contains an UAGGG sequence which is recognized by proteins in the HNRNPA1 family of RBPs. Thus, altering this sequence would be important for binding studies of these proteins.

More information would be helpful for the crosslinking step. Especially as two different conditions were used for array vs in solution UV crosslinking steps. What apparatus was used? How far away were the arrays from the UV bulbs? A supplementary figure could be helpful here.

What was the ratio of acrylamide:bis-acrylamide used for the 12% PAGE gels?

It would be good to mention more detail or include citations for psoralen crosslinking. How efficient is it? What is crosslinked? Should this be optimized in terms of crosslinking conditions or RNA linker sequence?

Supplementary Material:

A major issue is that the main text does not refer to Figures/Tables/Methods presented in the Supplementary Materials. Additionally, there is considerable interesting information (e.g. RNase H assay) “buried” in the Supplementary Materials section that could be mentioned and/or integrated with the main text. It is unfair that readers and reviewers should be expected to comb through it all with such minimal guidance.

In addition, the microarray designs (actual sequences) should be available through GEO or another source. The Agilent microarray summary in Supplementary Table 7 (which should be presented much earlier in the Supplementary Methods) is not adequate. Additionally, in Table 7, it is not clear what the “tc” abbreviation refers to. This should be made clear in the Table footnote.

Reviewer #3 (Remarks to the Author):

In this work, Schaudy and coworkers describe a new method for converting DNA arrays into an RNA product. High-complexity RNA arrays are important for investigating the relationship between RNA sequence, structure, and function and are a powerful tool for RNA engineering applications. In the past several years, various technologies that facilitate the synthesis of RNA arrays have been developed, however these approaches typically require expertise and instrumentation that is not readily available to most laboratories. Schuady et al. have developed a new approach for converting DNA arrays into RNA by photocrosslinking an RNA primer to a synthetic ssDNA template for use in a T7 RNAP primer extension reaction. After degrading the template DNA, the RNA remains attached to the array substrate due to the cross-linked primer. Overall, this is a creative method that has the potential to make RNA array-based biochemical assays more broadly accessible. I recommend that the manuscript be accepted after the comments below are addressed.

Comments:

1. The supplemental methods sections 'Enzymatic degradation of DNA template and determination of optimal linker length' and 'RNase H assay' (and the relevant Figures) would be more appropriately placed in the Results section of the main text. Although these sections are included as supplemental methods, they are primarily written as additional results sections and contain several crucial controls and validation assays. Given their importance for establishing the conditions used to degrade DNA after transcription without releasing the RNA, the authors should consider moving these sections to the Results, where they would be more readily visible. At minimum, these sections should be discussed and referenced at relevant points in the main text (see comment 4).

2. In Fig. 2a, each of the samples in lanes 1-3 were treated with TURBO DNase. It would likely be useful to perform this experiment alongside control reactions that were not treated with DNase, since this would allow for a more accurate assessment of the identity of each band.

3. Is 70% yield (relative to incorporation of 1 nt) of full length RNA sufficient for most RNA-array based assays? Were any variations of the in vitro transcription reaction conditions tested to try and optimize for a larger fraction of full length RNA? If not, this seems like a step of the procedure that would benefit from an attempt at optimization. For example, it may be worthwhile to test whether the full length RNA yield can be improved by additives like ssDNA binding proteins that enhance polymerase processivity (e.g. ET SSB or T4 Gene 32 protein, which are both commercially available), or formamide, which can alleviate single-stranded nucleic acid structure at low concentrations that tend not to inhibit enzymes. It would probably be fairly straightforward to test a few transcription conditions using the solution-based assay in Fig 2a to optimize for full length RNA yield and, if there is an apparent improvement, test the same conditions using the assay in Fig 2b.

4. None of the supplemental figures or tables are referenced in the main text. The authors should reference the supplemental figures at relevant locations in the Results and Methods sections.

5. Many of the supplemental methods sections essentially repeat sections of the main methods with slightly more detail. Could the related sections be merged together?

Minor comments:

1. 'Methods' is listed as a heading twice, once before 'sequence design' and once before 'Primer extension and template degradation'.

2. Assuming this is in line with the formatting guidelines, the supplemental materials might be easier to read if the supplemental tables were placed together after the figures and text.

REVIEWER COMMENTS

Reviewer #1 (Remarks to the Author):

While this is an interesting topic and interesting work, and the technical quality of the study seems fine, the paper is severely remiss in one important regard.

The basic idea presented in this paper has been published previously in 2014 by another group in the authors cited reference 15. Reference 15 is cited in a somewhat inscrutable manner in the introduction as follows: "Similar sequence complexity can be achieved by combining chemical DNA synthesis and enzymatic fabrication using nucleic acid photolithography¹⁵, or also enzymatically, but at lower density, via capture of RNA from DNA template arrays¹⁶. "

In fact the procedure using in reference 15 is conceptually identical to that used in the present paper, with the elements of a photolithographically fabricated DNA array being copied into RNA by T7 DNA polymerase, followed by enzymatic destruction of the DNA to leave only the RNA on the array. This fact was not recognized in the paper. The differences are in the details of the chemistry employed; it may be that the authors implementation is more practical and useful to the world, so I am not saying the result is without value - but it is not conceptually new, and probably does not belong in Nature Communications for that reason. I think the authors need to rewrite the paper to correctly position it relative to the work in reference 15 and then submit to a somewhat less prestigious venue with a transparent acknowledgement of its follow-on nature but some reasonable motivation for the work in its practical utility to the community.

Author's reply: Certainly it was not our intention to obscure the value of reference 15 (25 in the revised manuscript) through inscrutability. Like the other enzymatic methods described in references 11 to 16 (now refs. 21 to 28), the approach of the Smith group is quite complex and difficult to do justice to without a fair amount of text and a figure. In particular, their approach requires highly specialized chemical synthesis which, worldwide, only they (and perhaps we), are able to accomplish. Although our proposal to carry out this research was funded before ref. 15 was published, we studied their approach in great detail and it was very useful to our own progress. And in particular influenced our choice to use the T7 RNAP instead of the engineered TGK polymerase of Cozens et al (ref. 48). We did not cite ref 15 in the context of the use of the T7 RNA polymerase because the discovery of primer extension by this polymerase was actually made by Daube and von Hippel (old ref. 20, new 32). We now cite ref. 15 a second time in the context of the use of 2'-O-methyl RNA as primers for the T7 RNAP.

Regarding the value of more practical or widely usable approaches and whether they should be only published in lower tier journals, we would like to point out that science in general and molecular biology in particular owe much of their rapid advance to the introduction of approaches that democratize and/or greatly expand the throughput of experimental approaches. Gene editing, for example was available long before CRISPR-Cas9 was discovered, but is still important precisely because it makes gene editing widely accessible. Without

claiming our approach is similarly important, our approach makes RNA microarray synthesis universally available rather than just available in a few specially equipped labs.

Reviewer #2 (Remarks to the Author):

Summary:

Consistent with the title, this paper essentially describes a means to convert single-stranded oligonucleotides, often synthesized on arrays in which the DNA is coupled to the surface at its 3' end, into an equivalent array of RNA. This is done by first hybridizing a primer containing the T7 polymerase promoter sequence. The primer contains 5'-psoralen 2'-O-methyl, which can be crosslinked to the DNA strand. The T7 polymerase then transcribes the remainder of the oligonucleotide, then the DNA strand is degraded by TURBO DNase, leaving only single-stranded RNA with the complimentary sequence.

The technique is of obvious utility and would be of general interest as it represents a new method in general molecular biology. It may in fact be transformative for specialists who study RNA binding proteins, RNA folding, etc.

Author's reply: We thank the reviewer for his or her very careful reading of the manuscript, strong words of support for our approach, and constructive and detailed comments on how to improve the manuscript. We have endeavored to make the recommended changes and believe that the manuscript benefits strongly from the revisions.

The manuscript itself would benefit from a great deal of revision, so that others can actually use the technique and understand how it works.

The main text reads reasonably well, but that is because it is something of an advertisement. Exploration of many critical experimental parameters is relegated to the Supplementary document, and there is almost no connection between the main text and the Supplementary. The Supplementary Figures and Tables are not referenced or discussed in the main text, and there is little to no integration between the two. In addition, the Supplementary is very difficult to follow. There is no obvious overall organization, there are missing axis labels and no description of samples shown, etc. It also contains "data not shown" numerous times, which is inappropriate given that there are typically no space constraints on the Supplementary document.

As a consequence, the uninitiated reader (even a supposed expert in the field) is left with a great deal of decoding and inference. Without better presentation and integration of the Supplementary, it is hard to know what is scientifically most important. As this paper is about a method, it really must describe how the method really works, how it was developed, and how the critical parameters were explored. From the length of the Supplementary, I infer that that there has in fact been considerable exploration of parameter space.

I have enumerated a number of specifics below, which may not be comprehensive. I would suggest that much of the Supplementary should be moved to the main text. As an example, the length and sequence of the oligo-dT linker appears to be a critical parameter, and indeed, it is surprising that it is not cut by the DNase – presumably due to steric incompatibility with the array surface? It is claimed that its length was optimized, and I can see that there are some kinds of experiments in the

Supplementary, but I cannot decipher how they constitute optimization. There must be a tradeoff between efficiency of extension and prevention of DNase cleavage.

Author's reply: We originally took advantage of the Journal's policy allowing manuscripts to be submitted without formatting/structuring according to the Journal's standard until after review. In particular, we had written the manuscript as a brief communication, with most of the details in the SI. In this revision, we have integrated all the key result previously in the SI, as well as the supplementary methods, into the main text. We hope the article now presents the approach in a clearer, more comprehensive manner, including details of the exploration of the parameter space associated with the conversion to RNA. These details are now described in a new Figure 2 as well and several paragraphs of new text. For instance, linker length and chemistry were indeed varied in order to study how far does the DNase process and degrade the template strand. As can now be seen in Fig. 2a (and explained in the main text in the relevant section ("Determination of RNA identity and process optimization"), hybridization to the RNA following template degradation decreases in efficiency for linker lengths ≥ 10 -15nt, suggesting that the linker itself becomes long enough to be a substrate for the enzyme and is the reason why we settled for a 5-nt long linker.

Note that I am not proposing additional experiments. Only that what has been done needs to be presented in a more sensible and easily digested fashion. Questions below are simply what came to mind and if the answer is known it should be included.

Also, I should mention that these are running notes – I struggled to prioritize, and instead I am offering these comments as a starting point for overall reorganization of the paper.

Introduction:

The first paragraph is devoid of references and is overly general – there is no need to recite common knowledge. The comparison between DNA and RNA doesn't seem relevant and, the way it is presented, seems to suggest that DNA-target interactions don't generally involve hydrogen bonding, van der Waals forces, etc. which I don't think was the intent.

Author's reply: We have revised the first paragraph of the introduction to hopefully address these concerns as well as to add some references.

For the 2nd sentence, "These interactions are essential in all aspects of gene expression from splicing and post-transcriptional regulation to localization and translation ...", could be more accurately phrased as, "... essential in all aspects of post transcriptional gene regulation from splicing to localization and translation ..."

Author's reply: We have adopted the suggested wording.

2nd Paragraph: Citations 4 and 5 seem incorrect. Both studies analyzed RNA, primarily. However, the paper that DNA is being analyzed (e.g. "DNA pull-down arrays", etc.). And, as far as I recall, neither citation used SELEX.

Author's reply: We agree that those references were incorrect and have replaced them with Jolma et al 2010 and Gold et al 2010.

3rd Paragraph: Please list specific techniques being referenced (e.g. HiTS-RAP, RNA-MaP, etc.). The 2nd sentence which attempts to summarize references 11-14 is unclear. Also, terms such as “highest throughput” or “ultra-high-throughput” (and “very high throughput” — in the Abstract) are not meaningful.

Author’s reply: We have extended this section with the use of these terms as well as with a short description of how the RNA is generated in these approaches. We agree that in the absence of well-defined meaning for differences in density or throughput these terms are meaningless and have consolidated them all to just “high”.

Also, one potentially confusing aspect is use of the term “array”, which is commonly used in the literature to refer to microarrays. However, some of the high-throughput sequencing platforms cited in the text are being referred to as “arrays”. This is technically correct but nevertheless can create ambiguity. To avoid confusion, I would suggest using the full name for microarray (ie avoid abbreviating microarray to “array”) and/or terms such as “chip arrays” (for microarrays), “flow cell arrays” (for the high-throughput sequencing based systems), etc.

Author’s reply: We agree that the terms “array” and “microarray” are somewhat ambiguous and unsatisfactory. In our defense, the term arrays seems more favored these days, e.g. Buenrostro et al also use the term “RNA array” repeatedly to describe their RNA-MaP RNA output. However, we agree that using the terms “microarrays” and “flow cell microarrays” makes sense in the present context to distinguish between the two, and have modified both the title and text with this standard.

Results:

The two sections, “Enzymatic conversion from DNA arrays” and “Conversion to RNA of a commercial DNA array”, could easily be consolidated into one (e.g. “Conversion of DNA to RNA microarrays”).

Author’s reply: We have adapted our results section to consolidate these two sub-sections into one section.

There is very little text corresponding to Figures 1b, 1c or 1d. Specifically, there is one sentence saying an Agilent and a photolithographic were converted from DNA to RNA arrays. Which is followed by an entire paragraph on customizable Agilent arrays which seems better suited for the Introduction. Also, what is the relevance of these figures? Integration with the Supplementary Materials would be of benefit here. Importantly, it is not clear what the authors are trying to present in Figure 1d and Supplementary Figure 5.

Author’s reply: We have greatly expanded the Figure 1 legend to address these concerns. Regarding Figure 1d, the intent is to illustrate the high degree of conversion complexity and uniformity that can be achieved, something harder to see in the Agilent array conversion since there is no easy way to know what the random pattern of dots should look like, although we have tried to give some sense of this in an excerpt in Figure 1b. We have also added a few sentences (1st full paragraph on page 4) on the length of the heteroduplex in the elongation complex of the T7 RNAP and refer to the crystal structure shown in this figure. The length of this RNA:DNA duplex is relevant to the choice of primer length.

The Figure Legend for Fig 1 reads, “Enzymatic conversion of DNA arrays to RNA. (a) Three-steps for converting an existing DNA array to RNA. (b) Conversion of an Agilent SurePrint DNA microarray to RNA.” — clearly, the text is repetitive and could be improved. The sentence, “The second array has an identical layout but has been converted to RNA to allow hybridization to surface bound RNA sequences” seems incorrect. Should it say something like, “to allow hybridization to Cy3-labeled oligo, as in the first sub-array”? Also, the 1st and 2nd sub-arrays are boxed in blue/red — this should be mentioned in the Figure Legend.

Author’s reply: We have tried to fix the redundancy by shortening the start of (a). The wording “The second array has an identical layout but has been converted to RNA to allow hybridization to surface bound RNA sequences” has been optimized to hopefully remove this ambiguity: “The second subarray (outlined in orange) has an identical layout but has been converted to RNA, allowing hybridization to surface-bound RNA sequences with the same Cy3-labelled oligonucleotide”.

In Figure 1b, there is a microarray image with red and blue spots. It is unclear what this is — I assume it is a false-colour overlay, but I cannot find a description in the Figure Legend or main text.

Author’s reply: It is a legend describing the identity of the sequences in the Agilent microarray close-up. We have added this detail to the figure caption.

For Figure 1b, the spot intensities in the converted Agilent RNA array have substantial variability. The distribution of the median fluorescent spot intensities across the Agilent array should be shown.

Author’s reply: The variability in the spot intensity of the Agilent array is due to the presence of many different sequences, with variation in melting temperatures in both the template sequence and in the primer complement, resulting in signal intensity differences upon hybridization to the RNA product. The sequence information for the Agilent array is given in the supplementary information (Supplementary Table 1 and 3). This is also one of the reasons for including Figure 1d, as it allows a quick visual assessment of the uniformity, since spatial patterns in the efficiency of conversion to RNA would be visible as clear distortions in the image.

For the Cy3-UTP incorporation in panel 2b (photolithographic array), dA template positions greater than 40 have decreased fluorescent intensities which indicates a reduction in full-length RNA. This is paralleled by the smaller products (~43-50 nts) derived in solution in Figure 2a — presumably, these are RNA (?). How can these issues be explained and remedied? Perhaps the DNA to RNA conversion is not efficient as expected, or RNA detection using the Cy3-labeled oligo hybridization is suboptimal? On-array RNA conversion of DNA using Cy3-UTP (in Agilent arrays) could help address this issue. Is it possible that the DNA template sequences are truncated or is the T7 RNA polymerase prone to prematurely terminating from certain templates?

Author’s reply: The Cy3-UTP incorporation and the corresponding decrease in fluorescence with increasing length, we believe is mostly due to an artifact of Cy3 fluorescence itself, which is known to be dependent on the distance to the surface of the array, with close proximity increasing the fluorescence properties of the dye. We have, for instance, noted that hybridization signals decrease with increasing linker lengths with up to 40% weaker fluorescence on 25-nt long linker relative to a 5-nt long linker. We think that a substantial amount of the decrease in intensity seen in the graph in this figure (now Fig. 3b) is due to this increasing distance from the surface rather than a reduction of the full-length product. We

have added the following text to the top of page 5: “The decrease of fluorescent signal intensities with increasing distance of the fluorophore from the glass surface is consistent with previous observations upon hybridization of Cy3-labeled oligonucleotides to surface-bound DNA⁴⁷ (see also Supplementary Fig. 3).” While we can quantify this effect via hybridization base experiments, in the case of enzymatic labeling with Cy3-NTPs it is difficult to distinguish between fluorescence effects due to distance from the surface and possible enzymatic inefficiencies that may also be related to linker length.

Assuming that the yield of full length RNA is actually sub-optimal, it can probably be increased with longer incubation times. The trick would be to find a balance between the hydrolytic loss of the silane surface functionalization due to the warm water, as well as RNA degradation, and less RNA truncation. Again, our current thinking is that the loss in fluorescence observed in the Cy3-UTP experiments are mostly due to an artifact of fluorescence, but there is certainly room to improve the enzymatic process. Finally, just by the nature of primer extension, it seems likely that if T7 RNAP were prematurely terminated, it would still leave a (longer) primer that could then be further extended.

Again, I do not know what is the point of Figure 1d, it does not appear to show any meaningful data or information. The structures are from PDB. It is unusual to re-present data from another paper without appropriate context.

Author’s reply: As mentioned above, using an interpretable image allows the reader to easily judge the uniformity of conversion to RNA as well as well as the achievable complexity. The actual PDB structure seems relevant since it is the structure of the same T7 RNA polymerase elongation complex responsible for its ability to create the RNA arrays. We have also added a few sentences (1st full paragraph on page 4) on the length of the heteroduplex in the elongation complex of the T7 RNAP and refer to the crystal structure shown in this figure.

For Figure 2a, it would be helpful to have illustrations corresponding to the products analyzed by PAGE. For Lane 3, there are 2 two prominent smaller bands. Are these RNA or products from DNA digestion? This issue could be addressed via RNase digestion.

Author’s reply: We have added these illustrations of the products to Figure 2 (now Figure 3).

For Figure 2b, there is significant background fluorescence spots throughout the microarray. Of course, this would impact quantification and a downstream analysis. How is this issue addressed?

Author’s reply: Cy3 binds readily to glass unless it is attached to an oligo, which helps wash it away. We also observe this high background when chemically coupling Cy3 on an array using a Cy3 phosphoramidite. The bright spots within that background are probably aggregates of the Cy3-UTP. We account for this background by including control spots on the array that have the same sequence but lack the primer complementary section. Without primer hybridization there is no polymerization and therefore no incorporation of Cy3-UTP. We used the fluorescent signal from these spots for background subtraction. This background and background correction approach are described in the text: Methods\Cy3-UTP-based detection.

Methods:

There are two different “Methods” sections. I assume this was an accidental insertion of the second “Methods” header.

Author’s reply: We apologize for missing the presence of the second occurrence of “Methods”

The methods for microarray “Sequence design” only discuss dT length. No other information about the Agilent microarray sequences is given. An AMADID number is not provided as far as I am aware. Additionally, four different fluorescently labeled probes were used — all of which have different lengths and sequences. It is not clear how these sequences were selected or where they bind.

Author’s reply: We have added the Agilent AMADID number and more detailed information on the probes in the supplementary information. Nevertheless, it is unclear to us how useful this information will be since the important details, the optimal linker length and perhaps the primer sequence, were already provided. The remaining sequence designs (mainly the template sequences) are largely arbitrary and based allowing us to reuse labeled oligonucleotides we have previously used in unrelated experimental contexts.

More information regarding bias and/or efficiency during RNA primer extension via T7 RNA polymerase as well as comparison between RNA primer- and T7 promoter-mediated RNA transcription would be useful information to include in the manuscript.

Author’s reply: It would indeed be interesting, particularly from an enzymology perspective, but in this manuscript we are only concerned with optimizing the efficiency of the primer promoted mechanism since the promoter mediated mechanism plays no role in our experiments. Similarly, comparisons between primer extension with T7 RNA polymerase and the T7K polymerase (ref. 48) could also be quite useful, but are beyond the scope of the present manuscript.

As the dT and RNA linker remain after chemical/enzymatic treatment of microarrays, what specific optimizations could researchers use to prevent background and/or undesirable interactions (e.g. during RNA-RNA or RNA-protein interaction experiments, for example) in these regions?

Author’s reply: The DNA linker is already quite short, 5 to 10 nt, and seems unlikely to be a factor in most experiments. It can also be made shorter as shown in the (new) Figure 2c, as short as a single dT. It probably does not need to be a dT linker either if e.g. there is concern about hybridization with a rA homopolymer in the product RNA array. Non-nucleosidic linkers such as hexaethyleneglycol (HEG) can also be used as shown in Figure 2, but since DNA microarrays with such linkers are not commercially available, we have not elaborated on their use. The primer may be of greater concern because it is longer than the linker. It could be an advantage in that many/most RNA-protein interaction combinatorial experiments use constant RNA regions on both sides of the permuted section and the primer could be chosen to match the sequence of the desired 5’ constant region of the desired RNA product.

What rules govern sequence selection for the RNA linker? Are there considerations for length and/or sequence that the readers should be aware of? For example, the RNA linker contains an UAGGG sequence which is recognized by proteins in the HNRNPA1 family of RBPs. Thus, altering this sequence would be important for binding studies of these proteins.

Author's reply: Exploring the sequence landscape of the primer is difficult on microarrays because of the need for a separate 5'-psoralen oligonucleotide for each test primer sequence and the need to account for melting temperature differences and hybridization efficiencies. We actually used two different methyl RNA primers in these experiments:

- (1) 5'-Ps-UAGGGACACGGCGAA and
- (2) 5'-Ps-UAGACCAGGGUGGUUCAUGAUGAUGAC

The primer we used in all the array experiments (1) is a modified version of that chosen by Daube and von Hippel, with the same 12 nt 3' end plus the 5' Ps-UAG. The extra G helps with the melting temperature and the UA (along with their complement on the DNA strand) are optimal substrates for the psoralen crosslinking reaction. The other (2) was used in the gel experiments and is a modified (5' Ps-UA added) version of an oligo that we regularly use as a quality control hybridization oligo for testing microarray synthesis and to label fiducial features on arrays (with a 5' Cy3). Since neither was chosen with any special sequence features other than a reasonable T_m and minimum length, presumably almost any primer sequence will work. We have added a short discussion of primer choice in the first paragraph of page 4.

More information would be helpful for the crosslinking step. Especially as two different conditions were used for array vs in solution UV crosslinking steps. What apparatus was used? How far away were the arrays from the UV bulbs? A supplementary figure could be helpful here.

Author's reply: We have added additional information in the SI, in the form of a figure showing the setup and a description of the approach in the caption. Mainly we relied on a compact lab-made setup "UV box" with a high-power 365nm UV LED, a light pipe for spatial homogenization of the light, and a simple power supply. The UV box provides a convenient holder for standard glass slide microarrays, allowing homogenous, reproducible exposure. The UV intensity is measured using a calibrated power meter. There are also commercial sources of inexpensive 356 nm sources for lab use which should also work well.

What was the ratio of acrylamide:bis-acrylamide used for the 12% PAGE gels?

Author's reply: We have added this information to the section Methods/ Polyacrylamide gel electrophoresis analysis:

"After briefly spinning down, they were loaded and run on a 12% denaturing polyacrylamide gel (7 M urea; 19:1 ratio acrylamide:bis-acrylamide) with 0.5× TBE as running buffer."

It would be good to mention more detail or include citations for psoralen crosslinking. How efficient is it? What is crosslinked? Should this be optimized in terms of crosslinking conditions or RNA linker sequence?

Author's reply: We have a previous publication in Nature Communications (Ref 21; Hölz et al 2019) that has the full details on the efficiency of crosslinking reactions on microarrays and we have updated the section Methods/ Hybridization and photocrosslinking to emphasize this reference as a source of the full details on these optimizations.

Supplementary Material:

A major issue is that the main text does not refer to Figures/Tables/Methods presented in the Supplementary Materials. Additionally, there is considerable interesting information (e.g. RNase H assay) "buried" in the Supplementary Materials section that could be mentioned and/or integrated

with the main text. It is unfair that readers and reviewers should be expected to comb through it all with such minimal guidance.

Author's reply: We appreciate the reviewer's interest in the full details of the experiments. We moved all of the supplementary methods and most of the data on the linker length, DNase assays and RNase H assays to the main manuscript, including the addition of a new Figure (new Figure 2) with data and schemes for the template degradation with DNase, as well as the RNase H assay. We hope that the organization of the manuscript is now more reader friendly.

In addition, the microarray designs (actual sequences) should be available through GEO or another source. The Agilent microarray summary in Supplementary Table 7 (which should be presented much earlier in the Supplementary Methods) is not adequate. Additionally, in Table 7, it is not clear what the "tc" abbreviation refers to. This should be made clear in the Table footnote.

Author's reply: We have included the template sequences in our photolithographic arrays as well as those in Agilent arrays relevant for the discussion in the tables in the supplementary information. In addition, we have included the AMADID number, which includes all the sequences of the Agilent microarray. Microarray depositories, including GEO, are not appropriate because our microarrays do not contain any gene expression data, or indeed, any biologically relevant sequences. The meaning of the "tc" abbreviation (template complement) has now been defined in the new Figure 2 as well as in the SI table footnote.

Reviewer #3 (Remarks to the Author):

In this work, Schuady and coworkers describe a new method for converting DNA arrays into an RNA product. High-complexity RNA arrays are important for investigating the relationship between RNA sequence, structure, and function and are a powerful tool for RNA engineering applications. In the past several years, various technologies that facilitate the synthesis of RNA arrays have been developed, however these approaches typically require expertise and instrumentation that is not readily available to most laboratories. Schuady et al. have developed a new approach for converting DNA arrays into RNA by photocrosslinking an RNA primer to a synthetic ssDNA template for use in a T7 RNAP primer extension reaction. After degrading the template DNA, the RNA remains attached to the array substrate due to the cross-linked primer. Overall, this is a creative method that has the potential to make RNA array-based biochemical assays more broadly accessible. I recommend that the manuscript be accepted after the comments below are addressed.

Author's reply: We thank reviewer #3 for the very positive assessment and for the careful reading of the manuscript. We have carefully revised the manuscript to adopt the recommendations of the reviewer.

Comments:

1. The supplemental methods sections 'Enzymatic degradation of DNA template and determination of optimal linker length' and 'RNase H assay' (and the relevant Figures) would be more appropriately placed in the Results section of the main text. Although these sections are included as supplemental methods, they are primarily written as additional results sections and contain several crucial controls and validation assays. Given their importance for establishing the conditions used to degrade DNA after transcription without releasing the RNA, the authors should consider moving these sections to the Results, where they would be more readily visible. At minimum, these sections should be

discussed and referenced at relevant points in the main text (see comment 4).

Author's reply: We have moved all of the relevant material to the main text and have also added a new figure (new figure 2) that shows the data regarding the optimal linker length in the context of DNase degradation, as well as the data on RNase H degradation of the RNA following the conversion to RNA.

2. In Fig. 2a, each of the samples in lanes 1-3 were treated with TURBO DNase. It would likely be useful to perform this experiment alongside control reactions that were not treated with DNase, since this would allow for a more accurate assessment of the identity of each band.

Author's reply: We did repeat the gel experiments as recommended by the reviewer but found that the band identity assessment was similarly difficult and therefore decided not to include the additional data. The main problem seems to be that it is a complex system with three types of nucleic acid (DNA, RNA and 2'OMeRNA), which are sometimes in the form of chimeras (the primer after primer extension), or conjugated with each other via the photocrosslinking reaction. Two enzymes are also present, both the T7 RNAP and the DNase, and the products of these enzymatic reactions are present to varying amounts at the different stages of the process. Also present is the psoralen, which is also bound to a range of DNA fragment lengths after DNase degradation. Nevertheless, it is clear from the gel image that the RNA product in lane 3 is the same as the simulated product (chemically synthesized and purified RNA-2'OMeRNA chimera with a 5' psoralen) shown in lane 1. Although we included the gel data to demonstrate primer extension in solution in a traditional experimental format, this data combined with the microarray data in the same figure show unambiguous primer extension to full length or very near full length.

3. Is 70% yield (relative to incorporation of 1 nt) of full length RNA sufficient for most RNA-array based assays? Were any variations of the *in vitro* transcription reaction conditions tested to try and optimize for a larger fraction of full length RNA? If not, this seems like a step of the procedure that would benefit from an attempt at optimization. For example, it may be worthwhile to test whether the full length RNA yield can be improved by additives like ssDNA binding proteins that enhance polymerase processivity (e.g. ET SSB or T4 Gene 32 protein, which are both commercially available), or formamide, which can alleviate single-stranded nucleic acid structure at low concentrations that tend not to inhibit enzymes. It would probably be fairly straightforward to test a few transcription conditions using the solution-based assay in Fig 2a to optimize for full length RNA yield and, if there is an apparent improvement, test the same conditions using the assay in Fig 2b.

Author's reply: As the reviewer notes, the 70% full length yield for the long oligos is relative to the yield of 1-mers. The overall yield of RNA relative to the original DNA is lower and there are losses due, e.g. templates with un-hybridized primers, missing crosslink, or degradation of the linker with the DNase. We would certainly expect this to be sufficient for most applications, but in the end it will depend on the signal to noise ratio of any given experiment. We would probably not choose to do any additional optimization *in vitro*, since on-array experiments usually allow a much faster exploration of parameter space. We do intend to further optimize the conversion process and use the approach described in the manuscript to explore the sequence space of aptamers. We thank the reviewer for the suggestion on the use of ss DNA binding proteins as we were not aware of their use in enhancing polymerase processivity. Perhaps surprisingly, the main source of low yield in many microarray experiments is purely chemical, the loss of DNA/RNA from the surface during incubation/hybridization due to hydrolysis of the silane functional layer that serves as an interface between the glass and the

DNA. We are working on new silane chemistry as well, in order to improve this, but those are experiments completely unrelated to the current manuscript.

4. None of the supplemental figures or tables are referenced in the main text. The authors should reference the supplemental figures at relevant locations in the Results and Methods sections.

Author's reply: We have now integrated all of the text in the supplementary material into the main text and have added references in the main text to all supplemental figures and tables.

5. Many of the supplemental methods sections essentially repeat sections of the main methods with slightly more detail. Could the related sections be merged together?

Author's reply: We have moved all the supplemental methods into the main methods to eliminate this redundancy. The supplementary material now consists only of supplementary figures and tables.

Minor comments:

1. 'Methods' is listed as a heading twice, once before 'sequence design' and once before 'Primer extension and template degradation'.

Author's reply: We apologize for this mistake and have fixed it.

2. Assuming this is in line with the formatting guidelines, the supplemental materials might be easier to read if the supplemental tables were placed together after the figures and text.

Author's reply: We agree that the supplementary material can be better organized. After moving much of the material to the main text, we consolidated the remaining figures to the beginning of the supplementary material, followed by the supplementary tables.

REVIEWERS' COMMENTS

Reviewer #1 (Remarks to the Author):

The authors have satisfactorily addressed my comments (reviewer 1). I will leave it to reviewer 2 to assess the response to their comments. From my perspective it would be OK to proceed to publication.

Reviewer #2 (Remarks to the Author):

The paper has been substantially revised, and is much improved.

The revised paper would benefit if the following issues with the presentation were addressed:

Intro (2nd paragraph):

" ... using a sequencing or DNA pull-down microarrays to identify strong nucleic acid ligands after selection cycles."

—> It does not seem that any of the references listed used DNA pull-downs. The words "DNA pull-down" could simply be removed.

Fig 1.

Please include the names of the probes used.

Fig 2.

This figure is confusing for several reasons:

- 1) The schematic is the 2nd panel
- 2) Grey arrows are sometimes used to represent DNase treatment and sometimes used for other conditions (e.g. hybridization or no treatment).
- 3) Probe names aren't given and the P, tc, t, Pc, etc. designations seem confusing and unnecessary
- 4) The figures make it seem that a 5-Cy3-DNA probe can anneal to transcribed RNA (when the template is still present)

For simplicity/clarity I would suggest the following:

- 1) Have the schematic first, as panel a
- 2) Change the arrow colour scheme as grey arrows are being used to show untreated AND DNase treated (throughout) AND "Hybridization" (panel b).
- 3) Include the names of the probes used instead of the variety of labeled items (i.e "t", "P", "tc", "Pc", "tc (DNA)", "tc (RNA)", "t (DNA)", ssDNA (P-tc)", "DNA(P-tc)", and "RNA (Pc-t)" abbreviations in Figure panels and the Figure legend.
- 4) In situations where the Cy3-oligo doesn't hybridize to the transcribed RNA-DNA duplex, put a red "X" through the probe to highlight this point. The same can be done when the template is degraded.

Finally, the explanation for showing the microarray image in Figure 1, relative to a histogram showing the intensity values, doesn't make sense. I certainly hope there aren't "spatial patterns in the efficiency of conversion to RNA" - it doesn't seem as if this would be necessary to illustrate in the main figures, at the expense of displaying actual numbers. The reader should be able to trust that such artifacts would not be included in a journal submission.

Reviewer #3 (Remarks to the Author):

The authors have made substantial improvements to the manuscript and addressed all of my comments. The method is a useful contribution to the field.

My only remaining comment is that it may be worthwhile to incorporate Supplementary Figure 1c from the original manuscript back into the SI. I realize that this figure was condensed into Fig. 2b in the revised manuscript, but I found the original useful for understanding how the experiments in 2a and 2c were performed.

Simple synthesis of massively parallel RNA microarrays via enzymatic conversion from DNA microarrays

REVIEWERS' COMMENTS

Reviewer #1 (Remarks to the Author):

The authors have satisfactorily addressed my comments (reviewer 1). I will leave it to reviewer 2 to assess the response to their comments. From my perspective it would be OK to proceed to publication.

Author's reply: The authors thank reviewer #1 for the constructive comments.

Reviewer #2 (Remarks to the Author):

The paper has been substantially revised, and is much improved.

Author's reply: The authors thank reviewer #2 for the further very careful reading and additional helpful suggestions for improvements.

The revised paper would benefit if the following issues with the presentation were addressed:

Intro (2nd paragraph):

"... using a sequencing or DNA pull-down microarrays to identify strong nucleic acid ligands after selection cycles." → It does not seem that any of the references listed used DNA pull-downs. The words "DNA pull-down" could simply be removed.

Author's reply: We have removed pull-down from this sentence.

Fig 1.

Please include the names of the probes used.

Author's reply: The number of probes (thousands) is too large to list. Partial lists are provided in Supplementary Tables 1 and the Supplementary Data File and the full list of probes is available via the AMADID 086693 design file available on the Agilent website.

Fig 2.

This figure is confusing for several reasons:

- 1) The schematic is the 2nd panel
- 2) Grey arrows are sometimes used to represent DNase treatment and sometimes used for other conditions (e.g. hybridization or no treatment).
- 3) Probe names aren't given and the P, tc, t, Pc, etc. designations seem confusing and unnecessary
- 4) The figures make it seem that a 5-Cy3-DNA probe can anneal to transcribed RNA (when the template is still present)

For simplicity/clarity I would suggest the following:

- 1) Have the schematic first, as panel a
- 2) Change the arrow colour scheme as grey arrows are being used to show untreated AND DNase treated (throughout) AND “Hybridization” (panel b).
- 3) Include the names of the probes used instead of the variety of labeled items (i.e. “t”, “p”, “tc”, “Pc”, “tc (DNA)”, “tc (RNA)”, “t (DNA)”, ssDNA (P-tc)”, “DNA(P-tc)”, and “RNA (Pc-t)” abbreviations in Figure panels and the Figure legend.
- 4) In situations where the Cy3-oligo doesn’t hybridize to the transcribed RNA-DNA duplex, put a red “X” through the probe to highlight this point. The same can be done when the template is degraded.

Author’s reply: We agree that the Figure 2 remains confusing for a variety of factors as described by the reviewer. We address these individual points as follows:

- 1) The schematic b) is between panels a) and c) because the left side of the panel refers to a) and the right side to c). To make this clear, we have introduced symbols in b) directing the reader’s attention, either to the right or to the left. We also changed the color scheme such that the reader can see that the blue, light brown and orange plots refer to the same color DNA or RNA strands in b), and the color scheme in the plots in c) now directs the reader’s attention to the oligonucleotide strands of the same color (purple) on the right side of the schematic in b).
- 2) We agree that the color scheme of the arrows can help with clarity and have adopted the reviewer’s idea. We have also clarified the meaning of the arrows in the caption.
- 3) The names of the probes are already in a panel in Figure 2, but we have made them more prominent. We also have added the full names to the figure caption.
- 4) We did not add a red “X” through the Cy3-labeled probe because it can hybridize even before template degradation by a small amount, presumably by partial RNA displacement. It is this signal that allows a normalization relative to the control with no degradation.

Finally, the explanation for showing the microarray image in Figure 1, relative to a histogram showing the intensity values, doesn't make sense. I certainly hope there aren't "spatial patterns in the efficiency of conversion to RNA" - it doesn't seem as if this would be necessary to illustrate in the main figures, at the expense of displaying actual numbers. The reader should be able to trust that such artifacts would not be included in a journal submission.

Author’s reply: Spatial patterns are unavoidable in microarrays and originate both in the synthesis and the downstream use, in this case enzymatic processing and hybridization. All these processes involve liquids moving across the surface and chemical gradients. This is the main reason that probes and their replicates in microarrays are randomized across the surface, to average out these effects. In the case of the Agilent arrays, one such artifact can be seen in form of the ring pattern of the spots in Fig.1a, which likely results from how the droplets of activated phosphoramidite dry on the surface. Relying only on numerical values for intensities can lead to a loss of artifact information, or results in larger error bars, which obscures their origin. We, and also Agilent, use various experimental approaches, including sequencing, to look for spatial patterns in microarray synthesis but to a very good first approximation the quality of the microarray can be judged visually, which is why it is important and relevant to show high quality images of microarrays, ideally, with non-

randomized patterns; human brains are very good at pattern recognition in images, but much less good at seeing these patterns in numbers.

Reviewer #3 (Remarks to the Author):

The authors have made substantial improvements to the manuscript and addressed all of my comments. The method is a useful contribution to the field.

Author's reply: The authors thank reviewer #3 for the constructive comments.

My only remaining comment is that it may be worthwhile to incorporate Supplementary Figure 1c from the original manuscript back into the SI. I realize that this figure was condensed into Fig. 2b in the revised manuscript, but I found the original useful for understanding how the experiments in 2a and 2c were performed.

Author's reply: We have reintroduced the original Supplementary Figure 1c into the Supplementary information as Supplementary Figure 3.